# Anti-Food Allergic Compounds from *Penicillium griseofulvum* MCCC 3A00225, a Deep-Sea-Derived Fungus

**DOI:** 10.3390/md19040224

**Published:** 2021-04-16

**Authors:** Cui-Ping Xing, Dan Chen, Chun-Lan Xie, Qingmei Liu, Tian-Hua Zhong, Zongze Shao, Guangming Liu, Lian-Zhong Luo, Xian-Wen Yang

**Affiliations:** 1Key Laboratory of Marine Biogenetic Resources, Third Institute of Oceanography, Ministry of Natural Resources,184 Daxue Road, Xiamen 361005, China; xingcuiping123@126.com (C.-P.X.); xiechunlanxx@163.com (C.-L.X.); zhongtianhua@tio.org.cn (T.-H.Z.); shaozongze@tio.org.cn (Z.S.); 2Fujian Universities and Colleges Engineering Research Center of Marine Biopharmaceutical Resources, Xiamen Medical College, 1999 Guankouzhong Road, Xiamen 361023, China; cd@xmmc.edu.cn; 3College of Food and Biological Engineering, Jimei University, 43 Yindou Road, Xiamen 361021, China; liuqingmei1229@163.com (Q.L.); gmliu@jmu.edu.cn (G.L.)

**Keywords:** deep-sea microorganism, fungus, *Penicillium griseofulvum*, anti-food allergy, fungal metabolites, marine natural products

## Abstract

Ten new (**1**–**10**) and 26 known (**11**–**36**) compounds were isolated from *Penicillium griseofulvum* MCCC 3A00225, a deep sea-derived fungus. The structures of the new compounds were determined by detailed analysis of the NMR and HRESIMS spectroscopic data. The absolute configurations were established by X-ray crystallography, Marfey’s method, and the ICD method. All isolates were tested for in vitro anti-food allergic bioactivities in immunoglobulin (Ig) E-mediated rat basophilic leukemia (RBL)-2H3 cells. Compound **13** significantly decreased the degranulation release with an IC_50_ value of 60.3 μM, compared to that of 91.6 μM of the positive control, loratadine.

## 1. Introduction

For the past decade, the trend to discover new compounds from marine microorganisms continues to rise [1], especially from marine fungi [2,3], which accounted for 68% of the reported new marine natural products in 2019 [4]. Of particular importance is the *Penicillium* species, which are recognized as the richest source for the discovery of biologically important and structurally unique secondary metabolites [5,6,7,8].

As our ongoing research for novel and bioactive secondary metabolites from the deep sea-derived microorganisms [8,9,10,11], the fungal strain *Penicillium griseofulvum* isolated from the Indian Ocean sediment was selected for a systematic chemical examination. As a result, five carotanes, four naphthalenes, and three viridicatol derivates were obtained [12,13]. A continuous study, however, led to the isolation of 10 new (Figure 1) and 26 known compounds. Herein, we report the isolation, structure elucidation, and biological activity of these compounds.

## 2. Results and Discussion

Compound **1** was isolated as a white powder. Its molecular formula was established as C_17_H_18_N_2_O_4_ according to the protonated molecule peak at *m/z* 337.1176 [M + Na]^+^ in its (+)−HRESIMS (High Resolution Electrospray Ionization Mass Spectroscopy) spectrum, requiring ten degrees of unsaturation. The ^1^H and ^13^C NMR spectroscopic data (Appendix A, Table 1) displayed 17 carbons, characteristics of one mono-substituted aromatic unit [*δ*_H_ 7.24 (1H, br t, *J* = 7.4 Hz, H-4), 7.33 (2H, dd, *J* = 7.8, 7.3 Hz, H-3, 5), 7.46 (2H, d, *J* = 7.8 Hz, H-2, 6); *δ*_C_ 127.5 (d × 2, C-2/C-6), 128.3 (d, C-4), 129.1 (d × 2, C-3/C-5), 143.2 (s, C-1)], one ortho-disubstituted benzene moiety [*δ*_H_ 7.16 (1H, td, *J* = 7.6, 1.0 Hz, H-5′), 7.47 (1H, td, *J* = 7.8, 1.5 Hz, H-4′), 7.60 (1H, dd, *J* = 7.8, 1.4 Hz, H-6′), 8.51 (1H, d, *J* = 8.1 Hz, H-3′); *δ*_C_ 122.4 (d, C-3′), 124.2 (s, C-1′), 124.7 (d, C-5′), 128.8 (d, C-6′), 132.7 (d, C-4′), 138.8 (s, C-2′)], one methyl [*δ*_H_ 2.89 (3H, s, 7′-NMe); *δ*_C_ 26.8 (q, 7′-NMe)], two oxygenated methines [*δ*_H_ 4.25 (1H, d, *J* = 2.3 Hz, H-8); 5.16 (1H, d, *J* = 2.0 Hz, H-7) *δ*_C_ 75.6 (d, C-7), 77.8 (d, C-8)], and two carbonyls [*δ*_C_ 171.3 (s, C-7′), 174.0 (s, C-9)]. In the ^1^H–^1^H COSY (Correlation Spectroscopy) spectrum, correlations of H-2 (H-6)/H-3 (H-5)/H-4, H-3′/H-4′/H-5′/H-6′, and H-7 (*δ*_H_ 5.16, d, *J* = 2.0 Hz)/H-8 (*δ*_H_ 4.25, d, *J* = 2.3 Hz) confirmed the two benzene units and deduced another fragment of C-7/C-8. By the HMBC (Heteronuclear Multiple-bond Correlation) correlations of H-7 (*δ*_H_ 5.16) to C-1/C-2/C-6/C-9 and H-6′ (*δ*_H_ 7.60)/7′-NMe (*δ*_H_ 2.89) to C-7′, **1** was then assigned a phenylpropionyl moiety and a benzamide groups (Figure 2). However, the limited HMBC correlations hindered the connection of these two fragments. Fortunately, crystals of 1 were obtained. By the single X-ray crystallography (Figure 3), the absolute configuration of **1** was then unambiguously assigned as 2-(2*R*,3*S*-dihydroxy-3-phenyl-propionylamino)-*N*-methyl-benzamide, and named penigrisamide.

Compound **2** was afforded as a colorless oil. The molecular formula C_19_H_21_N_3_O_4_ was deduced from (+)-HRESIMS data (*m/z* 378.1418 for [M + Na]^+^), indicative of eleven degrees of unsaturation. The ^1^H and ^13^C NMR spectroscopic data (Appendix A, Table 2) exhibited 19 carbons, including three methyl singlets (one oxygenated), two methylenes, seven methines (five olefinic), and seven non-protonated carbons (one carbonyl and two ketone groups). These signals were closely similar to those of aurantiomide C (11) [14], except that the terminal amino group in **11** was replaced by the methoxy unit (*δ*_C_ 52.2) in **2**. The assumption was confirmed by the HMBC correlation of 17-OMe (*δ*_H_ 3.46) to C-17 (*δ*_C_ 173.9). Accordingly, the structure of **2** was determined as 17-deamino-17-methoxylaurantiomide C, and named aurantiomoate C.

Compound **3** was obtained as a colorless oil. Its molecular formula was established as C_17_H_27_N_3_O_5_ on the basis of the protonated molecule peak at *m/z* 376.1841 [M + Na]^+^ in its (+)-HRESIMS spectrum, requiring six degrees of unsaturation. Diagnostic NMR data for 3 suggested the presence of a pyroglutamylleucinmethylester (**20**) [15]. Moreover, the ^1^H–^1^H COSY correlation of H_2_-4″ (*δ*_H_ 3.63 m)/H_2_-5″ (*δ*_H_ 2.02 m) and H_2_-6″ (*δ*_H_ 2.18 m, 2.00 m)/H-2″ (*δ*_H_ 4.47, dd, *J* = 8.4, 2.8 Hz), with HMBC correlations from H-2″ (*δ*_H_ 4.47, dd, *J* = 8.4, 2.8 Hz) to C-4″/C-5″, and H-6″ (*δ*_H_ 2.18 m, 2.00 m) to C-1″/C-4″, allowed for the presence of another pyroglutamyl moiety. The absolute configuration of **3** was determined by the hydrolysis and derivation using Marfey’s reagent, and Nα-(2,4-dinitro-5-fluorophenyl)-l-alaninamide (FDDA) derivatives were compared with the retention times of standard FDDA-amino acids (Figure 4). On the basis of the above evidences, **3** was then assigned as *N*,*N*-pyroglutamylleucinmethylester.

Compound **4** was obtained as a colorless oil. Its molecular formula was established as C_12_H_23_NO_4_ based on the sodium adduct ionic peak at *m/z* 268.1526 [M + Na]^+^ in its positive HRESIMS spectrum, requiring two degrees of unsaturation. Its ^1^H and ^13^C NMR spectra were very similar to those of pyroglutamylleucinmethylester (**20**) [15], except for a 2-hydroxy-3-methylbutanoyl unit instead of a pyroglutamyl moiety in **4**. This was confirmed by the ^1^H–^1^H COSY correlations of H_3_-4′ (*δ*_H_ 1.00, d, *J* = 7.0 Hz) and H_3_-5′ (*δ*_H_ 0.84, d, *J* = 6.8 Hz) via H-3′ (*δ*_H_ 2.07 m) to H-2′ (*δ*_H_ 3.86, d, *J* = 3.7 Hz). Via detailed analysis of the HMBC spectroscopic data and using Marfey’s method (Figure 5), the absolute configuration of **4** was then assigned as methyl-2*S*-hydroxy-3-methylbutanoyl-l-leucinate.

The molecular formula of **5** was established as C_24_H_32_O_7_ by the ion peak at *m/z* 455.2040 [M + Na]^+^ in its positive HRESIMS. The ^1^H and ^13^C NMR spectra exhibited 24 carbons, including three doublets and five singlet methyls, one methoxyl, four methines (two oxygenated and two olefinic), and eleven quaternary carbons (six olefinic and two carbonyl carbons). These signals were closely similar to those of penicyrone A [16] except that the hydroxy (*δ*_C_ 82.6) at the C-9 position in penicyrone A was replaced by the carbonyl (*δ*_C_ 202.8) in **5**. This was confirmed by the HMBC correlations from H-7 (*δ*_H_ 6.38, d, *J* = 1.4 Hz)/H-11(*δ*_H_ 6.28, d, *J* = 1.4 Hz)/H_3_-19 (1.67, s)/H_3_-20 (*δ*_H_ 2.02, d, *J* = 1.4 Hz) to *δ*_C_ 202.8. Accordingly, **5** was established to be 9-dehydroxy-9-oxopenicyrone A, and named verrucosidinol A.

Compound **6** presented a molecular formula of C_24_H_34_O_8_ by positive HRESIMS at *m/z* 473.2140 [M + Na]^+^. Comparison of the ^1^H and ^13^C NMR spectra of **6** with those of verrucosidinol (**25**) [17] showed they were very similar except that two olefinic carbons at C-4 and C-5 in **25** were replaced by an epoxy group in **6**. This was evidenced by the HMBC correlations from H_3_-16 (*δ*_H_ 1.84) to C-1/C-2/C-3, H_3_-17 (*δ*_H_ 1.61) to C-3/C-4/C-5, and H_3_-18 (*δ*_H_ 1.28) to C-5/C-6/C-7. Therefore, **6** was established as 4,5-dihydro-4,5-epoxyverrucosidinol, and named verrucosidinol B.

Compound **7** had a molecular formula C_17_H_14_O_6_ as assigned by its positive HRESIMS at *m/z* 337.0690 [M + Na]^+^. Its ^1^H and ^13^C NMR spectroscopic data greatly resembled those of helvafuranone [18] except for an additional hydroxy substituent at the C-8 position. By detailed analysis of its 1D and 2D NMR spectroscopic data, **7** was then established as 8-hydroxyhelvafuranone.

Compound **8** gave a molecular formula C_10_H_21_NO_3_ as deduced by the protonated molecule peak at *m/z* 202.1504 [M − H]^−^ in its negative HRESIMS spectrum. The ^1^H NMR spectrum exhibited one methyl doublet at *δ*_H_ 0.96 (3H, d, *J* = 6.2 Hz, H-10), and two methyl singlets at *δ*_H_ 1.12 (3H, s, H-9) and *δ*_H_ 1.16 (3H, s, H-8). The ^13^C and DEPT spectra revealed the presence of 10 carbons, including three methyls, two methylenes, three methines, and one oxygenated and one carbonyl non-protonated carbon. In the ^1^H–^1^H COSY spectrum, correlations were found of H-6 via H-5 to H-4/H-3 and of H-3 to H_3_-10/H-2. By the HMBC correlations of H_2_-2 (*δ*_H_ 2.24, dd, *J* = 12.8, 5.2 Hz; 1.98, m) to C-1/C-4/C-10, H-6 (*δ*_H_ 3.21, d, *J* = 9.5 Hz) to C-4/C-7, and H_3_-8 (*δ*_H_ 1.16, s)/H_3_-9 (*δ*_H_ 1.12, s) to C-6/C-7, the planar structure of **8** was then established. To determine the absolute configuration of C-6, a dimolybdenum tetraacetate [Mo_2_(OAc)_4_]-induced circular dichroism (ICD) experiment was employed. The ICD spectrum exhibited a positive Cotton effect at 310 nm (Figure 6). The sign of the diagnostic band at about 310 nm was correlated to the absolute configuration of the chiral centers in the 1,2-diol moiety. According to the rule proposed by Snatzke, the positive sign suggested a positive torsional angle for the O-C-C-O moiety. It was ascertained that the 6*R*-form could maintain the favored conformation in which the bulkyl moiety and O-C-C-O center stayed away from each other. Based on the above evidence, the structure of **8** was then designated as 6*R*,7-dihydroxy-3,7-dimethyloctanamide.

Compound **9** was obtained as a white powder. The molecular formula C_11_H_22_O_5_ was deduced from (+)-HRESIMS data at *m/z* 257.1237 ([M + Na]^+^), indicative of one degree of unsaturation. The ^1^H NMR spectrum showed a methyl at *δ*_H_ 1.14 (d, *J* = 6.2 Hz, H-10) and a methoxyl at *δ*_H_ 3.65 (s, H-11). The ^13^C NMR and DEPT (Distortionless Enhancement by Polarization Transfer) data displayed 11 carbons, including one methyl, one methoxyl, five methylenes, three methines, and one carbonyl. In the ^1^H–^1^H COSY spectrum, two isolated spin systems were observed as H_2_-2 (*δ*_H_ 2.49, 2.42)/H-3 (*δ*_H_ 3.79)/H_2_-4 (*δ*_H_ 1.22)/H_2_-5 (*δ*_H_ 1.82) and H_2_-6 (*δ*_H_ 1.21)/H-7 (*δ*_H_ 3.54)/H_2_-8 (*δ*_H_ 1.61, 1.48)/H-9 (*δ*_H_ 3.94)/H_3_-10 (*δ*_H_ 1.14). These two fragments could be connected by the HMBC correlations of H_2_-2 (*δ*_H_ 2.49, 2.42) and H-11(*δ*_H_ 3.65) to C-1 (*δ*_C_ 173.7). Therefore, **9** was established as methyl-3,7,9-trihydroxydecanate.

Compound **10** was obtained as a colorless oil. Its molecular formula was established as C_10_H_18_O_4_ on the basis of the protonated molecule peak at *m/z* 225.1109 [M + Na]^+^ in its positive HRESIMS spectrum, requiring two degrees of unsaturation. The ^13^C NMR spectrum in association with the DEPT spectrum indicated 10 carbon signals ascribed to one methyl doublet (*δ*c 23.9, C-10), five *sp^3^* methylenes (*δ*c 42.8, C-2; 32.2, C-4; 24.6, C-5; 32.8, C-6; 46.5, C-8), three *sp^3^* methines (*δ*c 75.8, C-3; 76.2, C-7; 65.3, C-9), and one carbonyl (*δ*c 175.9, C-1). In the ^1^H–^1^H COSY spectrum, a long chain of C-2/C-3/C-4/C-5/C-6/C-7/C-8/C-9/C-10 could be deduced by correlations of H_2_-2 (*δ*_H_ 2.41)/H-3 (*δ*_H_ 3.76)/H_2_-4 (*δ*_H_ 1.64, 1.21)/H_2_-5 (*δ*_H_ 1.84, 1.59)/H_2_-6 (*δ*_H_ 1.52, 1.21)/H-7 (*δ*_H_ 3.57)/H_2_-8 (*δ*_H_ 1.48)/H-9 (*δ*_H_ 3.93)/H_3_-10 (*δ*_H_ 1.13). In the HMBC spectrum, H-3 (*δ*_H_ 3.76) was correlated to C-7 and C-1, which constructed a hexacyclic ring via an ether bond between C-1 and C-7. Accordingly, **10** was established as 9-hydroxy-3,7-epoxydecanoic acid.

By comparison of the NMR and MS data with those published in the literatures, 26 known compounds were determined to be aurantiomide C (**11**) [14], cyclopenin (**12**) [19], (−)-cyclopenol (**13**) [20], (3*S*)-1,4-benzodiazepine-2,5-diones (**14**) [21], 3-benzylidene-3,4-dihydro-4-methyl-lH-l,4-benzodiazepine-2,5-dione (**15**) [22], 3-methyl-3,4-dihydroquinazoline-4-one (**16**) [23], 1,2-dihydro-2,3-dimethyl4(3H)quinazolinone (**17**) [24], *N*,*N’*-1,2-phenylenebis-acetamide (**18**) [25], aconicarpyrazine B (**19**) [26], pyroglutamylleucinmethylester (**20**) [15], cyclo-(l-Trp-l-Phe) (**21**) [27], fructigenine A (**22**) [28], fructigenine B (**23**) [28], brevicompanine B (**24**) [29], verrucosidinol (**25**) [17], (*S*)-penipratynolene (**2****6**) [30], (*S*)-4-(2-hydroxybutynoxy)benzoic acid (**27**) [31], (*S*)-4-(2-hydroxybutoxy)benzoic acid (**28**) (CAS:1357392-03-0), (*S*)-2,4-dihydroxy-1-butyl(4- hydroxy)benzoate (**29**) [32], methyl *p*-hydroxybenzeneacetate (**30**) [33], 2-hydroxy phenyl acetic acid (**3****1**) [34], methyl homogentisate (**3****2**) [35], 5-hydroxymethyl-furaldehyde (**3****3**) [36], leptosphaerone A (**3****4**) [37], 3-methyl-2-penten-5-olide (**3****5**) [38], and (*R*)-mevalonolactone (**3****6**) [39].

All isolated compounds (**1**−**3****6**) were evaluated for their antifood allergic activities in RBL-2H3 cells. Compound **13** showed potent degranulation-inhibitory activity with an IC_50_ value of 60.3 μM, which was stronger than the commercially available antifood allergy medicine, loratadine (IC_50_ = 91.6 μM), while **14** and **29** showed weak effects with IC_50_ values of 167.0 and 134.0 μM, respectively (Table 3).

## 3. Materials and Methods

### 3.1. General Experimental Procedures and Fungal Fermentation

*Penicillium griseofulvum*, isolated from a sediment sample of the Indian Ocean at a depth of 1420 m, was deposited at the Marine Culture Collection of China (MCCC) with the accession number MCCC 3A00225. It was cultivated on corn medium in 100 × 1 L Erlenmeyer flasks for 62 days. The detailed general experimental procedures, fungal fermentation, and extraction were reported previously [12].

### 3.2. Isolation and Purification

The defatted extract (55.4 g) was separated by column chromatography (CC) over silica gel (500 g) using a CH_2_Cl_2_-MeOH gradient (0→100%, 49 mm × 460 mm) to give six fractions (Fr.1−Fr.6). Fr.2 (1.9 g) was subjected to ODS (octadecylsilyl) (H_2_O-MeOH, 5→100%, 15 × 460 mm, 0.5 L for each fraction) to attain five subfractions (sfrs) (sfrs.2.1–sfrs.2.5). Sfr.2.3 (155.0 mg) was purified by column chromatography on Sephadex LH-20 (100 g) (MeOH, 2.0 × 120 cm, 300 mL) to afford **26** (12.7 mg). Fr.3 (2.1 g) was subjected to column chromatography (CC) on ODS (70 g) (H_2_O-MeOH, 5→100%, 15 × 460 mm, 0.5 L for each fraction) to attain eleven subfractions (sfrs) (sfrs.3.1–sfrs.3.11). Sfr.3.3 (111.6 mg) was subjected to CC over Sephadex LH-20 (70 g) (MeOH, 2.0 × 120 cm, 300 mL) and silica gel (PE-EtOAc, 2:1, 17 × 305 mm) to yield **35** (25.7 mg). Sfr.3.5 (131.2 mg) was chromatographed on Sephadex LH-20 (100 g) (MeOH, 2.0 cm × 180 cm, 500 mL) resulting in two sub-subfractions (ssfrs) (ssfrs.3.5.1− ssfrs.3.5.2). Ssfr.3.5.1 (3.8 mg) was further purified by HPLC using gradient MeOH-H2O (20→70%, 10 × 250 mm, 4 mL/min) to provide **17** (2.4 mg). Ssfr.3.5.2 (41.0 mg) was purified using preparative TLC (CH_2_Cl_2_-Me2CO, 20:1) to give **16** (16.6 mg). Compound **15** (30.7 mg) was isolated from Sfr.3.6 (66.9 mg) by CC over Sephadex LH-20 (70 g) (MeOH, 2.0 × 120 cm, 300 mL). Sfr.3.8 (52.4 mg) was chromatographed on a Sephadex LH-20 (70 g) (MeOH, 2.0 × 120 cm, 300 mL) to give two sub-subfractions (ssfrs) (ssfrs.3.8.1− ssfrs.3.8.2), ssfrs.3.8.1 and ssfrs.3.8.2 were purified by preparative TLC on silica gel (CH_2_Cl_2_-MeOH, 20:1) to provide **24** (4.9 mg) and **9** (1.7 mg), respectively. Fr.4 (4.9 g) was subjected to ODS (130 g) (H_2_O-MeOH, 10→100%, 26 × 310 mm, 1.5 L for each fraction) to obtain twelve subfractions (sfrs) (sfrs.4.1− sfrs.4.12). Compound **12** (216.4 mg) was isolated from sfr.4.1 (304.0 mg) by CC over Sephadex LH-20 (100 g) (MeOH, 2.0 × 180 cm, 500 mL). Sfr.4.2 (906.0 mg) was chromatographed on a Sephadex LH-20 (225 g) column (MeOH, 3.5 × 180 cm, 800 mL) and silica gel (PE-EtOAc, 2:1, 46 × 457 mm) to yield **36** (152.9 mg). Sfr.4.3 (644.0 mg) was fractionated by CC over Sephadex LH-20 (225 g) (MeOH, 3.5 × 180 cm, 800 mL) to attain three sub-subfractions (ssfrs) (ssfrs.4.3.1–ssfrs.4.3.3), ssfr.4.3.3 (78.2 mg) was purified by Sephadex LH-20 (70 g) (MeOH, 2.0 × 120 cm, 200 mL), followed by preparative TLC (CH_2_Cl_2_- Me2CO, 10:1) to provide **33** (10.0 mg) and **34** (10.5 mg). Sfr.4.4 (270.9 mg) was subjected to CC over Sephadex LH-20 (100 g) (MeOH, 2.0 × 180 cm, 500 mL), further purified using preparative TLC (PE-EtOAc, 1:2) to obtain **14** (48.6 mg). Sfr.4.5 (33.3 mg) was purified by Sephadex LH-20 (70 g) (MeOH, 2.0 cm × 120 cm, 300 mL) to yield **18** (8.6 mg). Sfr.4.6 (270.9 mg) and sfr.4.7 (342.5 mg) were subjected to CC over Sephadex LH-20 (225 g) (MeOH, 3.5 × 180 cm, 800 mL) to attain **32** (4.0 mg) and **31** (2.9 mg), respectively. Sfr.4.9 and sfr.4.10 (376.6 mg) were fractionated by CC on Sephadex LH-20 (225 g) (MeOH, 3.5 × 180 cm, 800 mL) to obtain four sub-subfractions (ssfrs) (ssfrs.4.10.1–ssfrs.4.10.4). Ssfr.4.10.1 (191.0 mg) was subjected to Sephadex LH-20 (100 g) (MeOH, 2.0 × 180 cm, 500 mL) to attain **22** (117.2 mg), while **28** (3.5 mg) was isolated from ssfr.4.10.3 (10.3 mg) by preparative TLC (CH_2_Cl_2_-MeOH, 5:1). Sfr.4.12 (239.5 mg) was chromatographed on Sephadex LH-20 (100 g) (MeOH, 2.0 cm × 180 cm, 500 mL), further purified using preparative TLC (PE-EtOAc, 2:1) to yield **23** (46.2 mg). Fr.5 (40.0 g) separated by column chromatography (CC) over ODS (650 g) (H_2_O-MeOH, 5→80%, 49 × 460 mm, 3 L for each fraction) to obtain fifteen subfractions (sfrs.5.1–sfrs.5.15). Sfr.5.2 (1.7 g) was separated by CC over Sephadex LH-20 (225 g) (CH_2_Cl_2_-MeOH, 1:1, 3.5 × 180 cm, 1000 mL) to give three sub-subfractions (ssfrs) (ssfrs.5.2.1− ssfrs.5.2.3), ssfr.5.2.2 (126.0 mg) was subjected to Sephadex LH-20 (100 g) (MeOH, 2.0 × 180 cm, 500 mL), followed by preparative TLC (CH_2_Cl_2_-MeOH, 20:1) to provide **19** (2.9 mg). Ssfr.5.2.3 (103.0 mg) was purified by preparative TLC (CH_2_Cl_2_-MeOH, 10:1) to attain **29** (8.6 mg). Sfr.5.3 (625.0 mg) was subjected to CC over Sephadex LH-20 (225 g) (MeOH, 3.5 × 180 cm, 800 mL) to furnish five sub-subfractions (ssfrs) (ssfrs.5.3.1–ssfrs.5.3.5), ssfr.5.3.1(228.0 mg) was separated by silica gel (CH_2_Cl_2_-MeOH 50:1→10:1, 46 mm × 305 mm), then subjected to HPLC (MeOH-H_2_O, 55→65%, 10 × 250 mm, 5 mL/min) to yield **20** (22.8 mg). Compounds **27** (9.3 mg) and **30** (5.1 mg) were isolated from ssfr.5.3.3 (54.0 mg) and ssfr.5.3.5 (29.9 mg) by preparative TLC (CH_2_Cl_2_-MeOH, 20:1), respectively, while **10** (3.7 mg) was isolated from ssfr.5.3.4 (29.5 mg) by preparative TLC (EtOAc -MeOH, 50:1), and further purified by preparative TLC (CH_2_Cl_2_-MeOH, 20:1). Sfr.5.4 (3.3 g) was fractionated by CC over Sephadex LH-20 (225 g) (3.5 × 180 cm, CH_2_Cl_2_-MeOH 1:1, 1200 mL) to attain five sub-subfractions (ssfrs) (ssfrs.5.4.1−ssfrs.5.4.5), ssfr.5.4.2 (73.0 mg) was purified by preparative TLC (CH_2_Cl_2_-MeOH, 20:1) to provide **3** (11.0 mg). Ssfr.5.4.3 (1.6 g) was subjected to CC over Sephadex LH-20 (225 g) (3.5 × 180 cm, MeOH, 1200 mL) and preparative TLC (CH_2_Cl_2_-MeOH, 20:1) to yield **11** (29.4 mg). Sfr.5.5 (484.0 mg) was subjected to HPLC (MeOH-H_2_O, 20→40%, 10 × 250 mm, 5 mL/min), followed by preparative TLC on silica gel (CH_2_Cl_2_-MeOH, 10:1) to attain **7** (4.5 mg), **13** (34.1 mg), and **8** (4.2 mg). Sfr.5.7 (180 mg) was chromatographed on a Sephadex LH-20 (100 g) (MeOH, 2.0 × 180 cm, 500 mL), and then subjected to preparative TLC (CH_2_Cl_2_-MeOH, 20:1) to obtain **1** (1.5 mg). Sfr.5.11 (753.0 mg) was purified by CC over repeated Sephadex LH-20 (225 g) (MeOH, 3.5 × 180 cm, 800 mL) to obtain four sub-subfractions (ssfrs) (ssfrs.5.11.1- ssfrs.5.11.4), **21** (30.9 mg) was isolated from ssfr.5.11.2 (127.5 mg) by preparative TLC on silica gel using CH_2_Cl_2_-MeOH (10:1), while **5** (6.1 mg) was isolated from ssfr.5.11.3 (235.7 mg) by preparative TLC on silica gel (PE-EtOAc, 1:1). Sfr.5.12 (3.5 g) was separated by CC over SephadexLH-20 (CH_2_Cl_2_-MeOH, 1:1, 3.5 × 180 cm, 1200 mL) to attain three sub-subfractions (ssfrs) (ssfrs.5.12.1–ssfrs.5.12.3). Ssfr.5.12.1 (489.0 mg) was purified by Sephadex LH-20 (225 g) (MeOH, 3.5 × 180 cm, 800 mL) and silica gel (PE-EtOAc, 5:1→1:1, 46 × 305 mm), finally, by preparative TLC (CH_2_Cl_2_-MeOH, 10:1) to provide **6** (6.9 mg) and **25** (22.9 mg). Ssfr.5.12.2 (1.6 g) was purified by CC over repeated Sephadex LH-20 (225 g) (MeOH, 3.5 × 180 cm, 1000 mL) and preparative TLC (CH_2_Cl_2_-MeOH, 20:1) to yield **4** (3.1 mg) and **2** (24.6 mg).

Penigrisamide (**1**): Colorless needles; [α${]}_{D}^{25}$ +34.5 (c 0.20, MeOH); UV (MeOH) λ_max_ (log ε) 212 (3.03), 252 (2.77) nm; ECD (ACN) Δ*ε*_195_ +3.67, Δ*ε*_203_ +1.78, Δ*ε*_203_ +1.78, Δ*ε*_213_ +4.40, Δ*ε*_225_ −0.62, Δ*ε*_250_ +1.98; ^1^H and ^13^C NMR data, see Table 1; (+)-HRESIMS *m/z* 337.1176 [M + Na]^+^ (calculated for C_17_H_18_N_2_O_4_Na, 337.1164).

*Aurantiomoate C* (**2**): Colorless oil; [α${]}_{D}^{25}$ −20.8 (c 1.20, MeOH), [α${]}_{D}^{25}$+19.4 (c 1.20, CHCl_3_); UV (MeOH) λ_max_ (log ε) 211 (4.40), 305 (3.94) nm; ECD (ACN) Δ*ε*_191_ −20.6, Δ*ε*_228_ +10.7, Δ*ε*249 −7.66, Δ*ε*_272_ −1.60, Δ*ε*_294_ −2.81, Δ*ε*_330_ +2.17; ^1^H and ^13^C NMR data, see Table 2; (+)-HRESIMS *m/z* 378.1418 [M + Na]^+^ (calculated for C_19_H_21_N_3_O_4_Na, 378.1430).

5-Deoxypyroglutamyl-pyroglutamylleucinmethylester (**3**): colorless oil; [α${]}_{D}^{25}$−85.6 (c 0.27, MeOH); UV (MeOH) λ_max_ (log ε) 205 (3.77) nm; ECD (ACN) Δ*ε*_217_ +1.96, Δ*ε*_235_ −0.39, Δ*ε*_249_ +0.16; ^1^H and ^13^C NMR data, see Table 1; (+)-HRESIMS *m/z* 376.1841 [M + Na]^+^ (calculated for C_17_H_27_N_3_O_5_Na, 376.1848).

Methyl-2-hydroxy-3-methylbutanoyl-L-leucinate (**4**): colorless oil; [α${]}_{D}^{25}$ −42.9 (c 0.27, MeOH); UV (MeOH) λ_max_ (log ε) 203 (3.31) nm; ECD (ACN) Δ*ε*_210_ +0.98, Δ*ε*_234_ −0.11; ^1^H and ^13^C NMR data, see Table 1; (+)-HRESIMS *m/z* 268.1526 [M + Na]^+^ (calculated for C_12_H_23_NO_4_Na, 268.1525).

Verrucosidinol A (**5**): Colorless oil; [α${]}_{D}^{20}$ +86.8 (c 0.22, MeOH), [α${]}_{D}^{25}$ +82.7 (c 0.22, MeOH); UV (MeOH) λ_max_ (log ε) 205 (4.13), 231 (4.00), 298 (3.67) nm; ECD (ACN) Δ*ε*_187_ +1.57, Δ*ε*_205_ −7.27, Δ*ε*_296_ +7.91; ^1^H and ^13^C NMR data, see Table 2; (+)-HRESIMS *m/z* 455.2040 [M + Na]^+^ (calculated for C_24_H_32_O_7_Na, 455.2046).

Verrucosidinol B (**6**): Colorless oil; [α${]}_{D}^{20}$ + 32.3 (c 0.35, MeOH), [α${]}_{D}^{25}$ +34.6 (c 0.35, MeOH); UV (MeOH) λ_max_ (log ε) 240 (3.98) nm; ECD (ACN) Δ*ε*_195_ +0.66, Δ*ε*_214_ −0.89, Δ*ε*_254_ +4.30; ^1^H and ^13^C NMR data, see Table 2; (+)-HRESIMS *m/z* 473.2140 [M + Na]^+^ (calculated for C_24_H_34_O_8_Na, 473.2151).

8-Hydroxyhelvafuranone (**7**)*:* Colorless oil; [α${]}_{D}^{25}$ − 16.7 (c 0.03, MeOH); UV (MeOH) λ_max_ (log ε) 204 (4.28) nm; ECD (MeOH) Δ*ε*_193_ +2.23; ^1^H and ^13^C NMR data, see Table 2; (+)-HRESIMS *m/z* 337.0690 [M + Na]^+^ (calculated for C_17_H_14_O_6_Na, 337.0688).

6,7-Dihydroxy-3,7-dimethyloctanamide (**8**): Colorless oil; [α${]}_{D}^{25}$ −7.3 (c 0.15, MeOH); UV (MeOH) λ_max_ (log ε) 203 (3.09) nm; ECD (MeOH) Δ*ε*_225_ +0.02; ^1^H and ^13^C NMR data, see Table 1; (−)-HRESIMS *m/z* 202.1504 [M − H]^−^ (calculated for C_10_H_20_NO_3_, 202.1443).

Methyl-3,7,9-trihydroxydecanate (**9**): White powder; [α${]}_{D}^{20}$ −6.8 (c 0.19, MeOH), [α${]}_{D}^{20}$ −8.9 (c 0.19, CHCl_3_); UV (MeOH) λ_max_ (log ε) 205 (2.21) nm; ECD (MeOH) Δ*ε*_210_ +0.11; ^1^H and ^13^C NMR data, see Table 1; (+)-HRESIMS*m/z* 257.1237 [M + Na]^+^.

9-Hydroxy-3,7-epoxydecanoic acid (**10**): Colorless oil; [α${]}_{D}^{25}$ +15.7 (c 0.21, MeOH); UV (MeOH) λ_max_ (log ε) 205 (3.10) nm; ECD (MeOH) Δ*ε*_211_ +0.18; ^1^H and ^13^C NMR data, see Table 2; (+)-HRESIMS *m/z* 225.1109[M + Na]^+^ (calculated for C_10_H_18_O_4_Na, 225.1103).

### 3.3. X-ray Crystallography of **1**

Compound **1** was obtained as colorless needles from MeOH. Its crystallographic data were measured by an Xcalibur and Gemini single-crystal diffractometer with Cu Kα radiation (*λ* = 1.54184 Å). Space group P2_1_2_1_2_1_, a = 4.7555(2) Å, b = 14.7379(7) Å, c = 22.971(1) Å, *α* = *β* = *γ* = 90°, V = 1609.95(12) Å^3^, Z = 4, D_calcd_ = 1.371 mg/cm^3^; *µ* = 0.847 mm^−1^, F (000) = 704. The final R indicates *R* = 0.0484 (2682), w*R*_2_ = 0.1337 (3174). Crystallographic data of 1 have been deposited in the Cambridge Crystallographic Data Center, with deposition number 2072655. Copies of the data can be obtained, free of charge, on application to CCDC, 12 Union Road, Cambridge CB21EZ, U.K. (fax +44(0)-1233-336033; email: deposit@ccdc.cam.ac.uk).

### 3.4. Maryer’s Method

As reported [40], compounds **3** and **4** (each for 1 mg) were separately dissolved in HCl (1 mL) and incubated for 24 h. The hydrolysate was dried and dissolved in acetone. Then NaHCO_3_ and FDAA were added to incubate for 1 h. After being cooled, the mixture was dissolved in 50% aqueous CH_3_CN to yield FDDA derivatives. The corresponding standard amino acids were treated with the same procedures. The FDAA derivates were analyzed by HPLC at 254 and 340 nm by comparing the retention times with those of standards.

### 3.5. Induced CD (ICD) Experiment

Compound **8** and dimolybdenum tetracetate [Mo_2_(OAc)_4_] were resolved in dried DMSO. Their CD spectra were recorded immediately. Then the ICD spectra were measured every 3 min until they were stationary. The inherent CD data of compound **8** was subtracted to provide its induced CD spectrum as described previously [41,42].

### 3.6. Anti-Food Allergic Experiment

The in vitro anti-food allergic experiment was conducted according to the reported method [43]. Briefly, IgE-sensitized RBL-2H3 cells were treated with tested compounds for 1 h. Then cells were stimulated with dinitrophenyl-bovine serum albumin. The bioactivities were quantified by measuring the fluorescence intensity of the hydrolyzed substrate in an Infinite M200PRO fluorometer (Tecan, Zurich, Switzerland). Phosphate-buffered saline (PBS) buffer and loratadine were used as negative and positive controls, respectively.

## 4. Conclusions

From the deep sea-derived fungus *Penicillium griseofulvum* MCCC 3A00225, 10 new and 26 known compounds were obtained. The structures of the new compounds were determined by extensive analysis of their NMR and HRESIMS spectra, the absolute configurations were confirmed by different methods including the single X-ray crystallography, Marfey’s method, and ICD experiment etc. (−)-Cyclopenol (**13**) showed the strongest in vitro anti-food allergic activity with an IC_50_ value of 60.3 μM in IgE-mediated RBL-2H3 cells.

## Figures and Tables

**Figure 1 marinedrugs-19-00224-f001:**
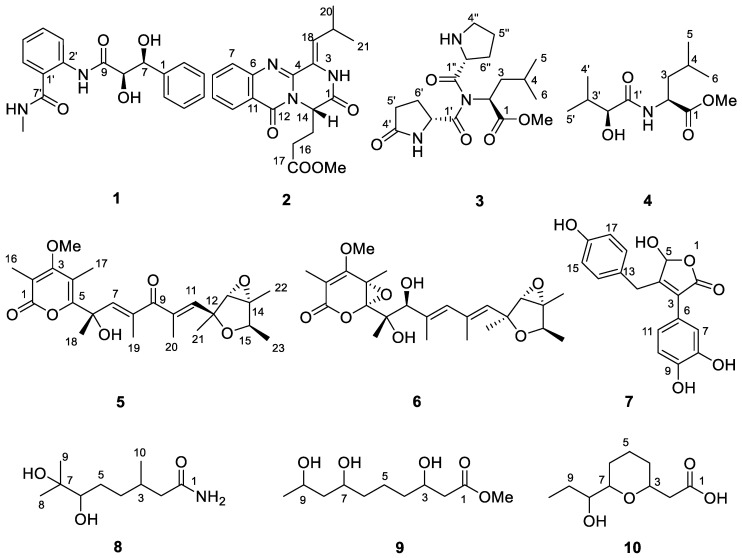
Compounds **1**–**10** from *Penicillium griseofulvum* MCCC 3A00225.

**Figure 2 marinedrugs-19-00224-f002:**
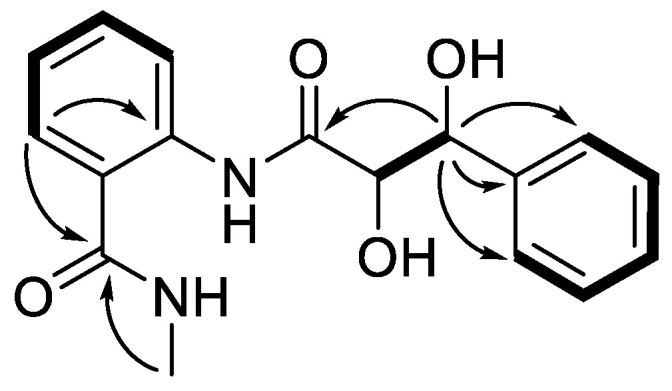
The key ^1^H–^1^H COSY (bold) and HMBC (arrow) correlations of **1**.

**Figure 3 marinedrugs-19-00224-f003:**
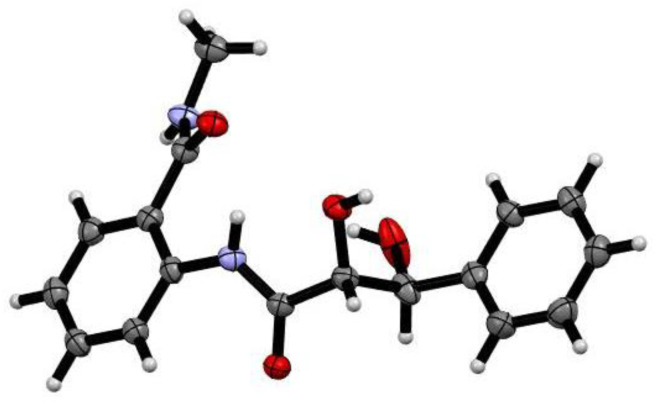
The X-ray crystallography of **1**.

**Figure 4 marinedrugs-19-00224-f004:**
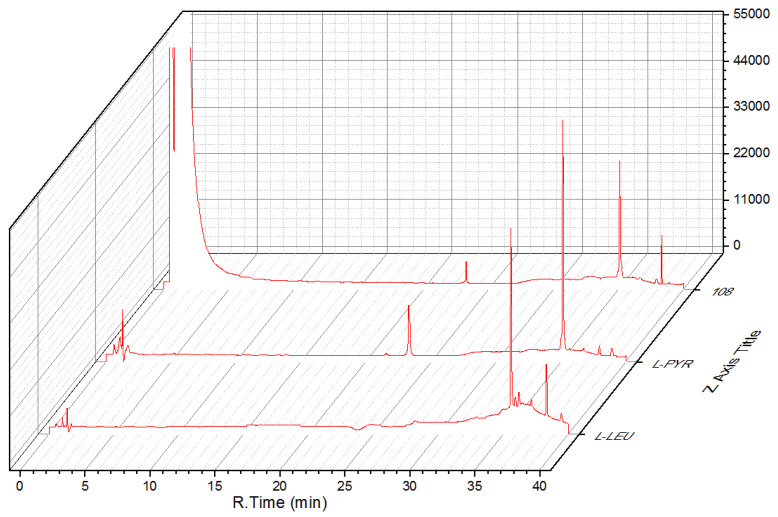
FDDA derivatives of **3** compared with the retention times of standard FDDA-amino acids.

**Figure 5 marinedrugs-19-00224-f005:**
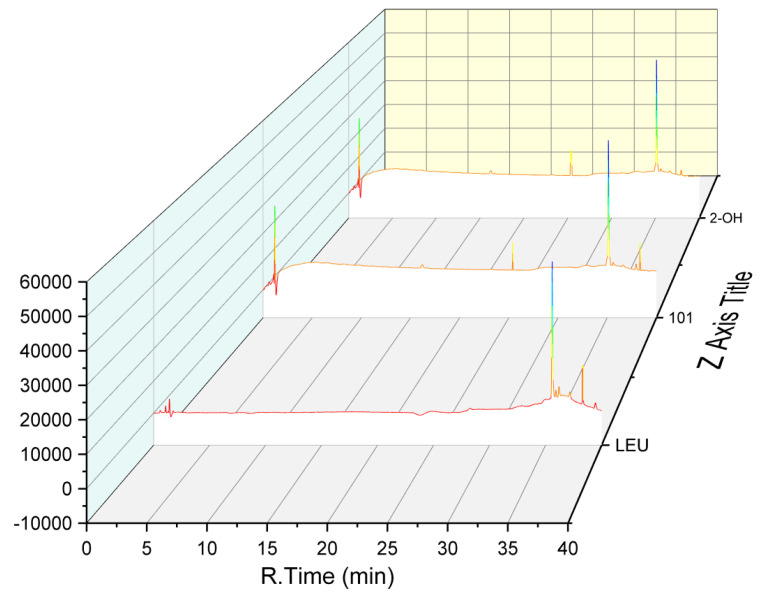
FDDA derivatives of **4** compared with the retention times of standard FDDA-amino acids.

**Figure 6 marinedrugs-19-00224-f006:**
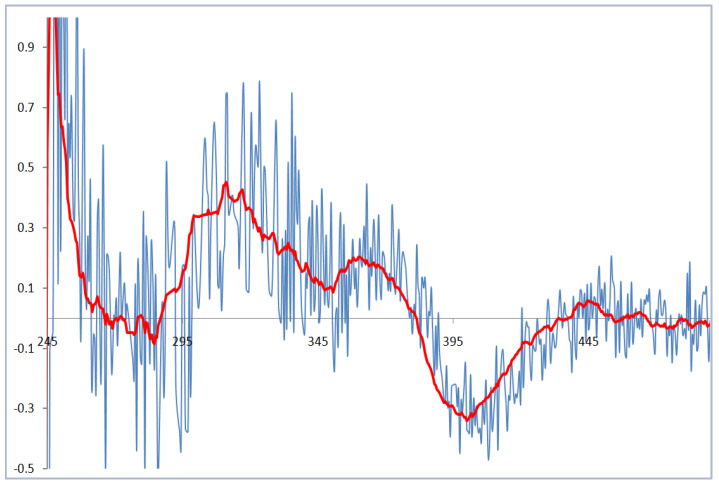
The induced CD spectrum of **8** in DMSO solution of Mo_2_(OAc)_4_.

**Table 1 marinedrugs-19-00224-t001:** ^1^H (400 MHz) and ^13^C (100 MHz) NMR spectroscopic data of **1**, **3**, **4**, **8**, and **9** in CD_3_OD.

No.	1		3		4		8		9	
	*δ* _C_	*δ* _H_	*δ* _C_	*δ* _H_	*δ* _C_	*δ* _H_	*δ* _C_	*δ* _H_	*δ* _C_	*δ* _H_
1	143.2 C		174.7 C		174.5 C		178.8 C		173.7 C	
2	127.5 CH	7.46 (d, 7.8)	52.3 CH	4.41 (dd, 8.9, 6.2)	51.6 CH	4.52 (dd, 9.8, 4.8)	44.0 CH_2_	2.24 (dd, 12.8, 5.2)1.98 m	42.2 CH_2_	2.49 (dd, 15.1, 4.3)2.42 (dd, 15.1, 8.9)
3	129.1 CH	7.33 (dd, 7.8, 7.3)	41.4 CH_2_	1.60 m	41.6 CH_2_	1.67 m	32.1 CH	1.94 m	75.9 CH	3.79 (tdd, 8.9, 4.4, 2.0)
4	128.3 CH	7.24 (br t, 7.4)	25.9 CH	1.74 m	26.0 CH	1.68 m	35.4 CH_2_	1.64 m; 1.22 m	32.1 CH_2_	1.62 m; 1.22 m
5	129.1 CH	7.33 (dd, 7.8, 7.3)	23.3 CH_3_	0.95 (d, 6.6)	23.3 CH_3_	0.95 (d, 6.2)	29.6 CH_2_	1.66 m; 1.23 m	24.3 CH_2_	1.82 m; 1.58 m
6	127.5 CH	7.46 (d, 7.8)	21.9 CH_3_	0.91 (d, 6.6)	21.7 CH_3_	0.92 (d, 6.2)	79.8 CH	3.21 (d, 9.5)	32.5 CH_2_	1.57 m; 1.21 m
7	75.6 CH	5.16 (d, 2.0)					73.8 C		78.5 CH	3.54 (tdd, 10.2, 3.7, 1.7)
8	77.8 CH	4.25 (d, 2.3)					25.8 CH_3_	1.16 s	45.9 CH_2_	1.61 m; 1.48 (dt, 14.0, 4.4)
9	174.0 C						24.8 CH_3_	1.12 s	67.4 CH	3.94 m
10							20.2 CH_3_	0.96 (d, 6.2)	23.1 CH_3_	1.14 (d, 6.2)
1′	124.2 C		172.9 C		176.7 C					
2′	138.8 C		56.2 CH	4.55 (dd, 8.8, 4.0)	77.0 CH	3.86 (d, 3.7)				
3′	122.4 CH	8.51 (d, 8.1)			33.0 CH	2.07 m				
4′	132.7 CH	7.47 (td, 7.8, 1.5)	181.6 C		19.5 CH_3_	1.00 (d, 7.0)				
5′	124.7 CH	7.16 (td, 7.6, 1.0)	30.3 CH_2_	2.36 m; 2.30 m	16.3 CH_3_	0.84 (d, 6.8)				
6′	128.8 CH	7.60 (dd, 7.8, 1.4)	25.5 CH_2_	2.47 m; 2.16 m						
7′ (1″)	171.3 C		174.4 C							
2″			61.3 CH	4.47 (dd, 8.4, 2.8)						
4″			48.1 CH_2_	3.63 m						
5″			25.9 CH_2_	2.02 m						
6″			30.3 CH_2_	2.18 m; 2.00 m						
NMe/OMe	26.8 CH_3_	2.89 s	52.6 CH_3_	3.69 s	52.7 CH_3_	3.70 s			52.1 CH_3_	3.65 s

**Table 2 marinedrugs-19-00224-t002:** ^1^H (400 MHz) and ^13^C (100 MHz) NMR spectroscopic data of **2**, **5**, **6**, **7**, and **10**.

No.	2 ^a^		5 ^a^		6 ^a^		7 ^b^		10 ^a^	
	*δ* _C_	*δ* _H_	*δ* _C_	*δ* _H_	*δ* _C_	*δ* _H_	*δ* _C_	*δ* _H_	*δ* _C_	δ_H_
1	167.7 C		167.1 C		167.9 C				175.9 C	
2			111.1 C		109.1 C		170.6 C		42.8 CH_2_	2.41 (d, 6.5)
3	126.3 C		171.2 C		168.3 C		120.3 C		75.8 CH	3.76, m
4	147.0 C		113.4 C		80.4 C		157.7 C		32.2 CH_2_	1.64, m; 1.21, m
5			160.8 C		109.1 C		96.5 CH	5.77 (d, 6.8)	24.6 CH_2_	1.84, m; 1.59, m
6	121.1 C		75.2 C		82.1 C		127.8 C		32.8 CH_2_	1.52, m; 1.21, m
7	128.4 CH	7.64 (d, 8.1)	147.7 CH	6.38 (d, 1.4)	90.5 CH	4.09 s	116.1 CH	6.92 (d, 1.8)	76.2 CH	3.57, m
8	136.0 CH	7.77 (td, 8.4, 1.4)	137.3 C		134.1 C		145.1 C		46.5 CH_2_	1.48, m
9	128.0 CH	7.46 (t, 7.6)	202.8 C		133.3 CH	5.84 s	146.1 C		65.3 CH	3.93, m
10	127.6 CH	8.14 (dd, 8, 1.1)	139.3 C		136.6 C		115.6 CH	6.79 (d, 8.2)	23.9 CH_3_	1.13 (d, 6.3)
11	148.7 C		143.7 CH	6.28 (d, 1.4)	133.2 CH	5.53 s	120.2 CH	6.75 (dd, 8.2, 1.8)		
12	162.1 C		81.3 C		81.6 C		31.0 CH_2_	3.85 (d, 15.1); 3.57 (d, 15.1)		
13			68.0 CH	3.64 s	68.7 CH	3.55 s	126.5 C			
14	56.2 CH	5.34 (t, 6.6)	68.7 C		68.7 C		129.6 CH	6.99 (d, 8.4)		
15	28.7 CH_2_	2.65 m; 2.15 m	78.5 CH	4.08 (dt, 6.8, 6.8)	78.3 CH	4.05 (d, 6.8)	115.5 CH	6.69 (d, 8.4)		
16	30.6 CH_2_	2.44 m	14.6 CH_3_	2.03 s	10.2 CH_3_	1.84 s	156.1 C			
17	173.9 C		11.1 CH_3_	2.09 s	20.8 CH_3_	1.61 s	115.5 CH	6.69 (d, 8.4)		
18	129.6 CH	6.33 (d, 10.4)	27.0 CH_3_	1.68 (d, 0.8)	19.7 CH_3_	1.28 s	129.6 CH	6.99 (d, 8.4)		
19	27.1 CH	2.87 m	13.4 CH_3_	1.67 s	15.8 CH_3_	1.86 (d, 0.9)				
20	22.5 CH_3_	1.13 (d, 6.6)	10.3 CH_3_	2.02 (d, 1.4)	19.1 CH_3_	1.93 s				
21	22.6 CH_3_	1.16 (d, 6.6)	21.1 CH_3_	1.38 s	22.1 CH_3_	1.37 s				
22			13.7 CH_3_	1.45 s	13.8 CH_3_	1.45 s				
23			19.3 CH_3_	1.17 (d, 6.8)	19.2 CH_3_	1.20 (d, 6.8)				
OMe	52.2 CH_3_	3.46 s	61.3 CH_3_	3.87 s	61.1 CH_3_	3.92 s				

^a^ CD_3_OD. ^b^ DMSO-*d*_6_.

**Table 3 marinedrugs-19-00224-t003:** Inhibition effects of compounds **1**–**36** on RBL-2H3 cell degranulation (*n* = 3).

Compound	IC_50_ (μM)
**13**	60.3
**14**	167.0
**29**	134.0
Others ^a^	≥ 200
Loratadine ^b^	91.6

^a^ Other compounds, including **1**–**12**, **15**–**28**, and **30**–**36**. ^b^ Loratadine was a commercially available anti-food allergic medicine.

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
