# Peer review of "Anti-Food Allergic Compounds from Penicillium griseofulvum MCCC 3A00225, a Deep-Sea-Derived Fungus"

_marinedrugs, 2021, doi:10.3390/md19040224_

Round 1

Reviewer 1 Report

This manuscript reports the isolation and characterization of 10 new and 26 known compounds from deep-sea-derived fungus Penicillium griseofulvum. Structure elucidation of new and known compounds was by analysis of spectroscopic data, as well as by X-ray analysis (for compound 1). The absolute configurations of some compounds were established by X-ray crystallography, Marfey’s method, and ICD method. In vitro anti-food allergic activity of the isolated compounds was evaluated, and one compound exhibited good activity. This work is useful for readers in the field, and it is therefore recommended for publication after minor revision. In order to improve this manuscript, please consider the comments and suggestions, which are listed below.

  1. “Keywords”; the word “Deep-sea” has broad meaning, so it should be “Deep-sea microorganism”. Keywords should contain the words “Fungal metabolites” and "Marine natural products".
  2. Please revise “was then undoubtedly assigned….” to “was then unambiguously assigned….”.
  3. Figure 4 and Figure 5 are too small to observe peaks for comparison. Please modify these Figures.
  4. “….confirmed by the COSY correlations of H3-4'…”; please use “1H-1H COSY”. Please correct this common issue throughout the manuscript.
  5. Please revise “(six olefinic 112 and two carbonyl)” to “(six olefinic 112 and two carbonyl carbons)”.
  6. “3.2. Fermentation and Purification”; please provide the method for fungal cultivation. Is it grown in PDB medium or rice medium? How many liters of fungal medium used for extraction and isolation of compounds?

Reviewer 2 Report

Review’s Comments: <Anti-Food Allergic Compounds from the Deep-Sea-Derived  Penicillium griseofulvum>

Minor error:

  • Comments: In line 53 spaces between ‘C-9)]. and In’
  • Comments: In line 56 mention the chemical shift of H3-8'
  • Comments: In the figure 1, indicate the position of 8', According to table 1, 8' is NMe.  Choose between 8' OR NMe?

Reviewer 3 Report

A carefull characterization of a set of new marine compounds.

Author Response

Thank you for your nice comments.